# Nanocomposites Based on Spin-Crossover Nanoparticles and Silica-Coated Gold Nanorods: A Nonlinear Optical Study

**DOI:** 10.3390/molecules28104200

**Published:** 2023-05-19

**Authors:** Eleni Zygouri, Aristeidis Stathis, Stelios Couris, Vassilis Tangoulis

**Affiliations:** 1Laboratory of Inorganic Chemistry, Department of Chemistry, University of Patras, 26504 Patras, Greece; eleni0503zig@hotmail.com; 2Department of Physics, University of Patras, 26504 Patras, Greece; a.stathis@iceht.forth.gr; 3Institute of Chemical Engineering Sciences (ICE-HT), Foundation for Research and Technology-Hellas (FORTH), 26504 Patras, Greece

**Keywords:** spin-crossover nanoparticle, gold nanorods, nonlinear optics

## Abstract

A nanocomposite based on silica-coated AuNRs with the aminated silica-covered spin-crossover nanoparticles (SCO NPs) of the 1D iron(II) coordination polymer with the formula [Fe(Htrz)_2_(trz)](BF_4_) is presented. For the synthesis of the SCO NPs, the reverse micelle method was used, while the gold nanorods (AuNRs) were prepared with the aspect ratio AR = 6.0 using the seeded-growth method and a binary surfactant mixture composed of cetyltrimethylammonium bromide (CTAB) and sodium oleate (NaOL). The final nanocomposite was prepared using the heteroaggregation method of combining different amounts of SCO NPs with the AuNRs. The nonlinear optical (NLO) properties of the hybrid AuNRs coated with different amounts of SCO NPs were studied in detail by means of the Z-scan technique, revealing that the third-order NLO properties of the AuNRs@SCO are dependent on the amount of SCO NPs grafted onto them. However, due to the resonant nature of the excitation, SCO-induced NLO switching was not observed.

## 1. Introduction

The switching between a paramagnetic high-spin (HS) state (S = 2) and a diamagnetic low-spin (LS) state (S = 0) using temperature as well as pressure and electromagnetic fields as external stimuli is a unique characteristic of Fe(II) spin-crossover (SCO) compounds [1,2,3,4]. This phenomenon of spin transition (ST) is accompanied by significant changes concerning the optical and magnetic properties of SCO compounds, as well as their chemical and physical characteristics, making them quite attractive for a plethora of applications [5,6,7,8,9,10,11,12]. Another interesting aspect of SCO compounds is the influence of their size and shape on the SCO properties, leading to novel processing methods for SCO materials at the nanometric scale [13,14,15,16,17,18,19]. A lot of attention has been given to the family of 1D coordination triazole-based Fe(II) compounds, as their SCO macroscopic properties are retained at the nanometric scale [20,21,22,23,24,25,26,27,28,29].

Recently, our group presented a novel nanosynthetic protocol [27] based on a reverse micelle method for the synthesis of water-soluble aminated silica hybrid SCO nanoparticles (NPs) of the 1D coordination polymer [Fe^II^(Htrz)_2_(trz)](BF_4_), where (Htrz = 1,2,4-1H-triazole), displaying stable aqueous and ethanol colloidal dispersions and retaining their SCO characteristics, accompanied by relevant thermochromic features (from colorless for the HS state to purple for the LS state). The novelty of this method is based on a two-step hydrolysis/condensation mechanism of tetraethyl orthosilicate (TEOS), in the first step, and an appropriate mixture of 3-aminopropyltriethoxy silane (APTES) with TEOS in the second step, leading to the final aminated silica hybrid SCO NPs.

Introducing functional groups onto the silica surfaces of SCO NPs and grafting luminescence molecules [30,31] and/or gold NPs [32,33,34,35,36,37] is a promising strategy for obtaining multifunctional nanoplatforms. Especially for the case of hybrid SCO@Au NPs, the photothermal heating properties of the AuNPs in both visible and near IR due to the nonradiative decay of surface plasmons significantly reduces the energy power responsible for the application of the SCO phenomenon [36,37]. Recently, it was found that nanocomposites of SCO@Au NPs, where the gold nanoparticles have the morphology of nanorods (AuNRs), can improve the diffusion of heat between the AuNRs and SCO NPs. The broad longitudinal surface plasmon resonance (LSPR) band in the near IR of AuNRs was found to be temperature-dependent upon the HS–LS switching of the SCO NPs [34]. A less pronounced shift of the SPR peak has been observed on hybrid thin films of Au@SCO and/or relative heterostructures [35,37].

In all the above studies, the scientific interest has been mainly focused on the monitoring of the SPR dependence on the SCO phenomenon, while the effects of the latter on the nonlinear optical (NLO) response of SCO@Au NPs have been left relatively unexplored so far. Having in mind that the experimental evidence of SCO-induced NLO switching in iron(II) complexes is still in its infancy [38], the need for more experimental studies and evidence aiming to correlate the SCO phenomenon with the NLO response is of the outmost importance for applications related to NLO-switching phenomena. A promising scenario to develop NLO devices/switches is the incorporation of suitable SCO materials with thermal hysteresis loops centered at room temperature, accompanied by NLO properties that are temperature sensitive.

From this point of view, in the present work, the influence of the SCO phenomenon on the NLO response of a nanocomposite consisting of silica AuNRs coated with aminated silica hybrid SCO NPs of the 1D iron(II) coordination polymer is investigated. For this purpose, a heteroaggregation method for the combination of the SCO NPs with the AuNRs was employed. AuNRs carrying different loads of SCO NPs were prepared and their NLO response was studied using the Z-scan technique [39]. For the Z-scan measurements, the visible (i.e., 532 nm) and infrared (i.e., 1064 nm) outputs of a 4 ns Q-switched Nd:YAG laser were selected for use because they allow for the resonant excitation of the transverse and longitudinal surface plasmon resonances (SPRs) of the AuNRs located at ~530 and ~1060 nm, respectively, allowing for their efficient photothermal heating.

## 2. Results and Discussion

### 2.1. General Synthetic Aspects

The synthesis and characterization of the aminated silica hybrid SCO NPs have been presented elsewhere [27], while the general synthetic protocol is also shown in Figure 1. In general, the reverse micellar method was used, according to which two water phases, (a) the metal salt of Fe(BF_4_)_2_.6H_2_O and (b) the ligand HTrz in a 1:3 molar ratio, were mixed with the organic phase consisting of the surfactant (Triton X-100) and co-surfactants n-hexanol and cyclohexane also playing the role of the organic solvent. To further functionalize the SCO NPs with aminated silica shells, a two-step hydrolysis/condensation protocol was followed, according to which (a) 100 μL TEOS was added in the two aqueous phases; (b) 100 μL APTES (3-aminopropyltriethoxy silane) and 100 μL TEOS were added after mixing the two water-in-oil microemulsions. According to the distributions obtained from TEM images [27], the size of the SCO NPs in EtOH is close to 45 nm, with a spherical morphology (Figure 1a), while aggregation phenomena appear in DMF solutions (Figure 1b).

A seed-mediated growth method was used for the synthesis of rod-like gold NPs using a mixture of surfactants (CTAB/NaOL) [40]. Absorption peaks at ca. 518 nm and ca. 1027 nm were revealed in the absorption spectrum of the AuNRs, denoted as transverse SPR (t-SPR) and longitudinal SPR (l-SPR), respectively. The aspect ratio (AR) of the AuNRs is directly related to the position of the l-SPR peak, according to Equation (1) [41]:(1)$$\lambda_{max}\left(nm\right)=95R+420$$
where *R* is the aspect ratio, and the calculated AR is close to 6.2.

According to the TEM images, the mean dimensions of the rod-shaped gold NPs are 90.0 ± 8.0 nm and 15.3 ± 3.4 nm concerning the length and width, respectively, while the aspect ratio of AR ~6.1 is close to the value obtained from the UV–VIS–NIR measurements (Figure 2 and Appendix A). The silica coating of the AuNRs succeeded with the hydrolysis/condensation of TEOS in a basic medium (pH = 10.5~11.0). The mesostructured growth of the silica was obtained using the CTAB/NaOL as a template, and under these highly basic conditions, both hydrolysis and condensation occur at the same time, leading to the almost vertical orientation of the mesoporosity of the silica shell to the gold core (Figure 2b,c) [42]. The effective full coating of the AuNRs with SiO_2_ was further supported by the fact that the colloidal stability of the core−shell nanoparticles was maintained in DMF solution, with no aggregation phenomena. The absorption spectra of the AuNRs@SiO_2_ dispersions present a t-SPR band at ca. 1057 nm, revealing a redshift of 30 nm due to the higher refractive index of the mesoporous SiO_2_, as shown in Figure 2e [43]. Furthermore, the effective coverage of the AuNRs with SiO_2_ is reflected in ζ-potential measurements, where the value of the ζ-potential was decreased considerably from high positive values of 36.7 mV (CTAB-covered AuNRs) to negative values close to −7.2 mV for AuNRs@SiO_2_ (Figure 2d).

A heteroaggregation procedure [44] was followed to coat the silica-covered AuNRs@SiO_2_ dispersed in DMF with aminated silica-covered SCO NPs dispersed in EtOH by simply mixing the two solutions. In this nonaqueous mixing procedure, the SCO NPs are destabilized in the DMF solution, assembling a dense coating on the surfaces of the AuNRs@SiO_2_. Three amounts of an ethanol solution of the SCO NPs were used (100 μL, 150 μL, and 200 μL) for mixing with the DMF solution of GNRs@SiO_2_, and the final colloidal dispersions in DMF were stable for months. It was found that for amounts greater than 200 μL, the resulting colloidal dispersion in DMF was unstable and quickly decolored. In all three cases, the AuNRs are surrounded by SCO NPs, creating aggregated patterns (Figure 3). This is probably because the SCO NPs are aggregated in DMF solution, with no well-defined morphologies (Figure 1c).

### 2.2. UV–Vis–NIR Absorption Measurements of AuNRs@SCO

The UV–Vis–NIR absorption spectra of the DMF solutions of AuNRs@SiO2 and AuNRs@SCO are presented in Figure 4. The transverse and longitudinal surface plasmon resonances (SPRs) of the AuNRs@SCO hybrids were located at 520 and 1060 nm, respectively. The absorption spectra of the AuNRs@SCO dispersions present a systematic small redshift of the l-SPR peak by 9 nm, compared to the respective AuNRs@SiO_2_ dispersions, attributed to the presence of the SCO NPs on their surfaces.

This redshift is commonly observed when iron NPs are deposited onto AuNRs [45,46], and it is attributed to the higher refractive index of iron NPs than that of mesoporous SiO_2_ (1.28–1.45) [47]. In addition, as the amount of SCO NPs was increased, a decrease in the absorbance was observed. Product loss during centrifugation could be responsible for this decrease [44].

### 2.3. Nonlinear Optical Properties of AuNRs@SCO

The NLO properties of the AuNRs@SCOs were systematically studied under resonant excitation conditions (i.e., by exciting them at the transverse (i.e., 532 nm) and longitudinal (i.e., 1064 nm) SPR peaks). In Figure 5, some representative OA and CA Z-scans of AuNRs@SCO dispersed in DMF are presented, under 4 ns and 532 and 1064 nm laser excitations, respectively while general information about Z-scan experimental setup and theory is presented in the Appendix A. It should be noted that the solvent (i.e., DMF) did not exhibit any NLO response for the range of laser intensities employed in the present experiments. In addition, the NLO response of the SCO NPs in ethanol/DMF dispersions was also investigated under identical experimental conditions to those used for the AuNRs@SCO dispersions. However, no significant NLO response was observed. Therefore, the Z-scan recordings of the AuNRs@SCO dispersions shown in Figure 5 directly reflect the NLO response of the AuNRs@SCO hybrid material. The symbols shown in Figure 6 correspond to the experimental data points, while the continuous lines correspond to the theoretical fitting of the OA and CA recordings by Appendix A, respectively. For the accurate determination of the NLO parameters of the AuNRs@SCOs, the Z-scan measurements were performed for a wide range of incident laser intensities, ranging from 10 to 65 MW cm^−2^. As can be seen from Figure 5a, the AuNRs@SCO dispersions were found to exhibit saturable absorption (SA) behavior under infrared (i.e., 1064 nm) laser excitation, corresponding to the negative nonlinear absorption coefficient (β). The observed SA behavior is attributed to the resonant excitation conditions met. In fact, the electric field of the resonant laser radiation induces the effective collective oscillation of the conduction-band electrons and the holes in the valence bands of the AuNRs. In general, excitation in the vicinity of the l-SPR results in the efficient excitation of the electrons from the valence to the conductive band, resulting in empty states of positive charges (i.e., holes) in the valence band accompanied by photoexcited electron–hole pairs, which are converted to hot carriers. Next, on a time scale of from 100 fs to 1 ps [48], the hot carriers redistribute their energy among carriers possessing lower energies through electron–electron scattering [49]. As a result, the hot carriers cool down via a relaxation process, building an equilibrium Fermi–Dirac-like distribution. As the sample approaches the focal plane (i.e., experiences higher laser intensity), the interband transitions become more efficient, causing the bleaching of the ground-state plasmon band, expressed as the saturation of the absorption (i.e., SA behavior) [50].

Under visible excitation (i.e., 532 nm), all the AuNRs@SCO samples exhibited reverse saturable absorption (RSA) (see, e.g., Figure 5c), corresponding to the positive nonlinear absorption coefficient (β). The observed RSA behavior can be described in terms of two-photon absorption (TPA) and/or excited-state absorption (ESA) [51]. In general, TPA is a relatively weak nonlinear process, enabled by virtual states, resulting in the relatively weak (or negligible) depopulation of the ground state, thus giving rise to an intensity-independent nonlinear absorption coefficient (β). On the contrary, ESA is a stronger process, as it involves real intermediate states, leading to the more efficient depletion of the ground state, giving rise to an intensity-dependent nonlinear absorption coefficient (β) [52]. In addition, as has been discussed elsewhere [51], ESA is more likely to occur under ns excitation. To shed more light on the mechanism responsible for the RSA behavior, Z-scan measurements were performed under different laser intensities, and the nonlinear absorption coefficient (β) of the AuNPs@SCOs was determined. The variation in the β values with the incident laser intensity is presented in Figure 6. As can be seen, the nonlinear absorption coefficient (β) is clearly intensity-dependent, suggesting that ESA is most probably the main mechanism responsible for the RSA behavior under 532 nm laser excitation.

As far as concerns the NLO refractive response of the AuNRs@SCOs, all the dispersions were found to exhibit self-defocusing behavior, as depicted in Figure 6b,d, corresponding to the negative nonlinear refractive parameter (*γ*′), as evidenced by the characteristic peak–valley configuration of the CA Z−scans. Some representative CA Z-scans, obtained under both excitation conditions (i.e., 532 and 1064 nm), are shown in Figure 5b,d. From these measurements, the *γ*′ was determined. Then, the nonlinear refractive index (*n*_2_) could be calculated from Equation (2):(2)$$n_{2}\left(esu\right)=\frac{cn_{0}}{40\pi }\gamma^{\prime }\left(\frac{m^{2}}{W}\right)$$
where *c* is the speed of light (in m s^−1^), and *n*_0_ is the linear refractive index at the laser excitation wavelength.

The determined values of the nonlinear absorption coefficient (β), nonlinear refractive index parameter (*γ*′), and nonlinear refractive index (*n*_2_) are listed in Table 1. The imaginary and real part of the third-order susceptibility (i.e., Imχ^(3)^ and Reχ^(3)^) were calculated using Appendix A, respectively. Finally, by inserting the values obtained from Appendix A into Appendix A, the magnitude of the nonlinear third-order susceptibility (χ^(3)^) was determined and is also listed in Table 1. To make it easier to compare the values of the NLO parameters of the AuNRs@SCOs with different loads of SCOs, the shown values have been normalized by the corresponding linear absorption coefficient (α_0_) at the respective laser excitation wavelength. Thus, they all refer to the absorption coefficient α_0_ = 1 cm^−1^. As can be seen from this table, the magnitudes of all the NLO parameters were found to increase with the load of SCO NPs. A graphical representation of this trend is presented in Figure 7. As can be seen from the plots in Figure 7, the variation in the NLO response of the AuNRs@SCOs (i.e., third-order susceptibility (χ^(3)^)) versus the SCO load is slightly larger in the case of infrared (i.e., at 1064 nm) excitation.

### 2.4. Investigation of SCO Phenomenon in AuNRs@SCO and Dependence of NLO Response on Spin Transition

The SPR dependence on the SCO phenomenon, for the amounts of 150 μL and 200 μL of SCO NPs in the final hybrid AuNRs@SCO, was investigated and is shown in Figure 8. To assess the influence of temperature on the samples, a specific methodology was employed. Initially, 300 μL of the sample was transfused into a 1 mm thick cuvette, which was then placed within a homemade copper sample holder wrapped with a constantan wire, allowing for its controlled heating via an ITC502S Oxford Instruments temperature controller. The temperature was stabilized within ±0.1 °C. Once the temperature reached ~80 °C, the cuvette was removed from the sample holder and the UV–VIS–NIR spectra were recorded. For both samples, a blue shift of the l-SPR appeared when the temperature of the DMF solution of AuNRs@SCO was at 80 °C. The value of the shift was determined to be about 20 and 30 nm for the cases of the 150 μL and 200 μL samples, respectively. This shift depends on the amount of SCO NPs and is related to the spin transition from LS to HS, as the refractive index in the LS state is higher than in the HS state. Quickly after taking the absorption spectra of the samples, to avoid the cooling of the solutions, measurements of the NLO response of the AuNRs@SCO samples were performed. However, the measurements revealed negligible changes in their NLO properties compared to those measured under room-temperature conditions. In fact, the determined NLO parameters (nonlinear absorption and nonlinear refractive index (*n*_2_)) were found to be almost identical (within the experimental accuracy) to those determined in the case of room-temperature samples. It should be noted at this point that because the laser excitation of the samples was performed at 1064 nm, full resonant excitation conditions were met. Although the applied heating of the samples resulted in a significant shift of the corresponding l-SPR peaks, the resonant excitation conditions are still valid, as can be seen from the enlarged view of the absorption spectra shown in Figure 8b,d. Thus, most probably, any change in the NLO properties of the samples arising from the SCO phenomenon cannot be observed due to the much stronger resonant excitation contributions, effectively preventing the observation of the expected much weaker NLO contribution due to the SCO phenomenon. Hence, it can be concluded that although heating can trigger the SCO phenomenon, its possible contribution to the NLO response of the hybrid samples is unobserved due to the resonant character of the excitation.

## 3. Materials and Methods

### 3.1. Material–Instrumentation–Physical Measurements

All manipulations were performed under aerobic conditions using reagents and solvents (Alfa Aesar, Haverhill, MA, USA; Sigma Aldrich, St. Louis, MO, USA; Serva, Catoosa, OK, USA) as received: HAuCl4·xH2O (Alfa Aesar, 99.999%, where x was estimated as 3); cetyltrimethylammonium bromide (CTAB) (Alfa Aesar, 99%); sodium oleate (NaOL) (Alfa Aesar, 99%); AgNO3 (Alfa Aesar, 99.9995%); ascorbic acid (Alfa Aesar, 99.5%); KBr (Alfa Aesar, ACS, 99% min); NaBH4 (Sigma-Aldrich, 99%). The ligands 1H-1,2,4-triazole (Htrz) and 4-amino-4H-1,2,4-triazole (NH_2_trz) and the tetraethyl orthosilicate (TEOS) and (3-aminopropyl)triethoxysilane (APTES) were purchased from Alfa Aesar, while the iron(II)tetrafluoroborate hexahydrate salt, Fe(BF_4_)_2_.6H_2_O, and n-hexanol were purchased from Sigma Aldrich. Triton X-100 and cyclohexane were obtained from Serva and were used without further purification. The deionized water used for synthesis was deoxygenated by simultaneous sonication and argon bubbling during 1 h.

The TEM study was performed utilizing an FEI CM20 TEM operating at 200 kV. TEM specimens were prepared by drop casting a 3 μL droplet of AuNRs@SCO nanoparticle suspension in DMF on a carbon-coated Cu TEM grid. The size of the particles was determined with “manual counting” using ImageJ software v 1.54d (https://imagej.net accessed on 10 April 2023). The UV–VIS–NIR measurements were conducted utilizing a double-beam Jasco V-670 spectrophotometer.

### 3.2. Synthesis of Gold Nanorods (AuNRs) and Silica-Covered AuNRs@SiO_2_

The synthetic protocol of the AuNRs is presented in detail elsewhere [53]. An amount of 75 μL of 0.1 Μ aqueous CTAB solution and 250 μL of 0.1 Μ aqueous NaOH solution were injected into 10 mL of the previously prepared AuNR solution (O.D. = 2.9, C[Au] = 260.6 μg/mL), which was kept at 29 °C to avoid CTAB crystallization and stirred for 5 min. A total of 120 μL TEOS was then added at a rate of 30 μL/30 min. The resulting solution was stirred for about 48 h. The silica-coated AuNRs were washed three times with H_2_O with centrifugation at 6000 rpm for 20 min, and the resulting pellet was redispersed in DMF.

### 3.3. Synthesis of Spin-Crossover Nanoparticles [Fe(Htrz)_2_(trz)](BF_4_).1.4SiO_2_.0.7H_2_O.0.4Acetone (**SCO**)

An aqueous solution of Fe(BF_4_)_2_·6H_2_O (337 mg, 1.00 mmol) in 0.5 mL of deionized H_2_O and 0.1 mL of TEOS was added to a solution containing Triton X-100 (1.8 mL), n-hexanol (1.8 mL), and cyclohexane (7.5 mL). The resulting mixture was stirred for 30 min until the formation of a clear water-in-oil microemulsion. A similar procedure was applied to 1,2,4,1H-Triazole (HTrz) (210 mg, 3.00 mmol) in 0.5 mL of deionized H_2_O. Both microemulsions were quickly combined, and the mixture was stirred for 24 h in the dark until the addition of 100 μL APTES. After 30 min of stirring, 100 μL TEOS was added, and the stirring continued for a further 24 h, followed by the addition of acetone to break the microemulsion. The precipitated nanoparticles were isolated by centrifugation at 6000 rpm, washed several times with EtOH/acetone, and finally dried under vacuum. Anal. Calcd. for **SCO**: C: 18.45; H: 2.54; N: 26.89%. Found: C: 18.65; H: 2.68; N: 26.50%. The final product can be dispersed in water and EtOH.

### 3.4. Synthesis of AuNRs@SiO_2_@SCO

For the mixing of GNRs@SiO_2_ with SCO NPs, two separate solutions were prepared: (a) a solution (1 mL, O.D. = 2.41, C[Au] = 199.5 μg/mL) of GNRs@SiO_2_ in DMF, and (b) an ethanol solution 1 mL (10 mg/mL) of SCO NPs. Injecting more than 200 μL of the ethanol solution of SCO NPs into the prepared DMF solution of GNRs@SiO_2_ resulted in a destabilized mixture that was discolored rather quickly. Therefore, three different solutions were prepared by mixing different amounts of the SCO NP ethanol solution ((1) 100 μL, (2) 150 μL, and (3) 200 μL) in the prepared DMF solution of GNRs@SiO_2_. The mixtures were treated in the ultrasound for about 15 min. The resulting solution was then centrifuged for 15 min at 5900 rpm. The supernatant solution was removed, and the pellet at the bottom of the falcon was redispersed in 1 mL DMF and kept at 4 °C. The final colloidal dispersion was stable for several months.

## 4. Conclusions

A facile method for the preparation of a nanocomposite based on silica-coated AuNRs with the aminated silica-covered spin-crossover nanoparticles (SCO NPs) of the 1D iron(II) coordination polymer with the formula [Fe(Htrz)_2_(trz)](BF_4_) is presented. The nonlinear optical (NLO) properties of the hybrid AuNRs coated with different amounts of SCO NPs were studied in detail by means of the Z-scan technique, revealing that the third-order NLO properties of the AuNRs@SCO are dependent on the amount of SCO NPs grafted onto them. The triggering of the SCO phenomenon was possible using a heating process, and a shift of the l-SPR peak (20–30 nm) was observed at 80 °C, related to the spin transition from LS to HS. However, changes in the NLO response were not observed, most probably masked by the stronger resonant NLO response, preventing their individual observation and quantification. A possible scenario for the effective monitoring of the NLO properties of SCO materials could be based on avoiding the thermal triggering of the SCO phenomenon and/or employing laser excitation, being nonresonant with the plasmonic features. These findings may pave the way for the development of new strategies for monitoring the SCO dependence of NLO properties for applications related to NLO-switching phenomena.

## Figures and Tables

**Figure 1 molecules-28-04200-f001:**
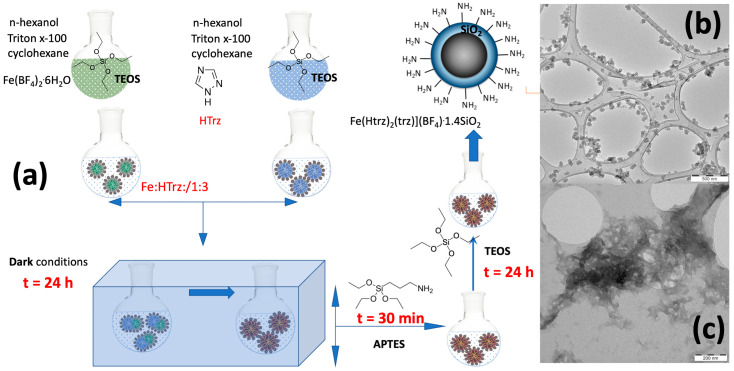
(**a**) Synthetic protocol for SCO NPs. (**b**) TEM images of SCO NPs in EtOH (white bar scale of 500 nm). (**c**) TEM images of SCO NPs in DMF (white bar scale of 200 nm). See text for more details.

**Figure 2 molecules-28-04200-f002:**
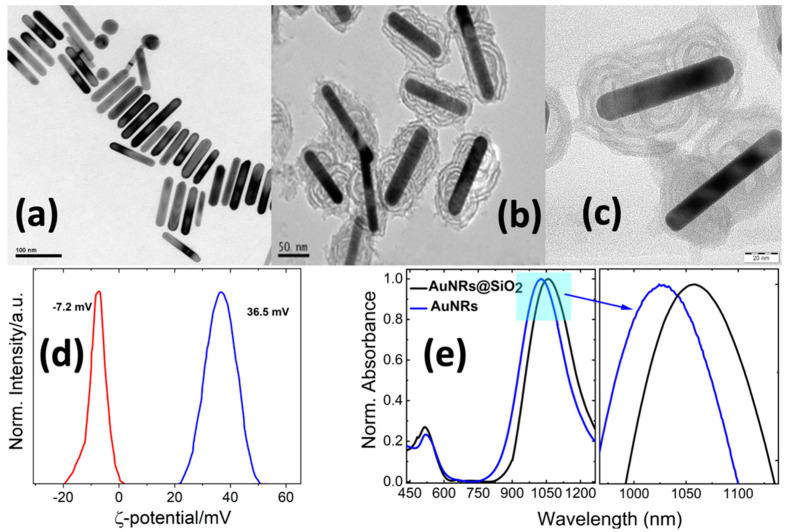
(**a**) TEM images of AuNRs in water. (**b**,**c**) TEM images of AuNRs@SiO2 in DMF. (**d**) ζ-potential measurements of AuNRs and AuNRs@SiO2. (**e**) UV−Vis−NIR absorption spectra of AuNRs and AuNRs@SiO_2_.

**Figure 3 molecules-28-04200-f003:**
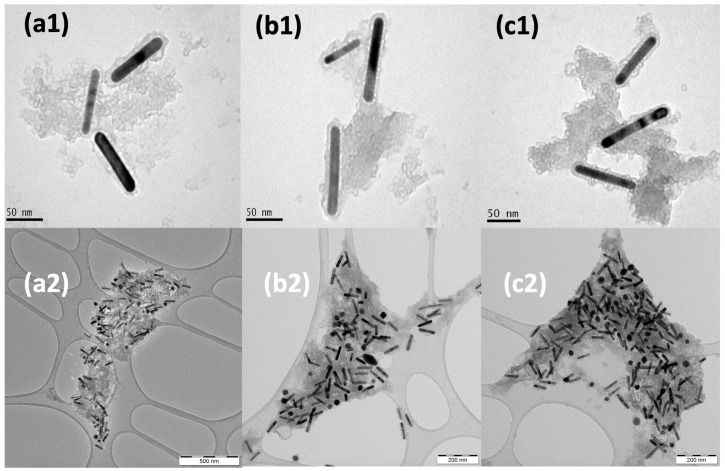
TEM images of AuNRs@SCO in DMF at different amounts of SCO NPs: (**a1**,**a2**) 100 μL, (**b1**,**b2**) 150 μL, and (**c1**,**c2**) 200 μL. The aggregation patterns of the upper line of images (**a1**–**c1**) involve a small number of AuNRs surrounded by SCO NPs, while in the bottom line of images (**a2**–**c2**), larger aggregated forms are displayed.

**Figure 4 molecules-28-04200-f004:**
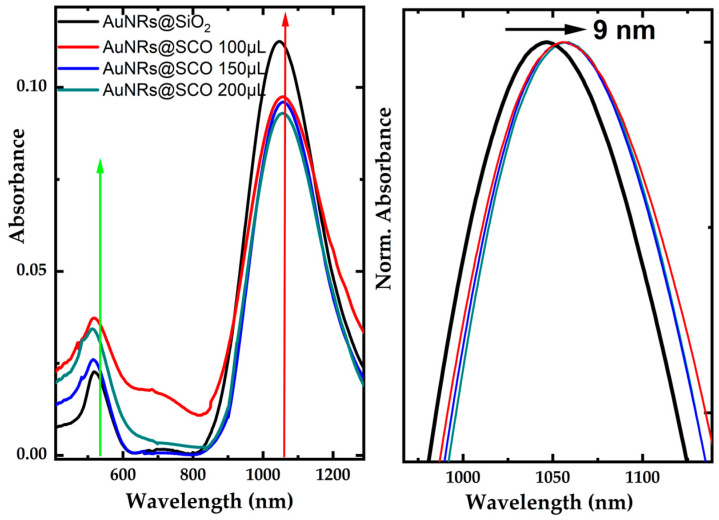
UV–Vis–NIR absorption spectra of AuNRs coated with different loads of SCO NPs (**left**). The arrows indicate the laser excitation wavelengths (i.e., 532 and 1064 nm). Normalized view of the l-SPR peaks (**right**).

**Figure 5 molecules-28-04200-f005:**
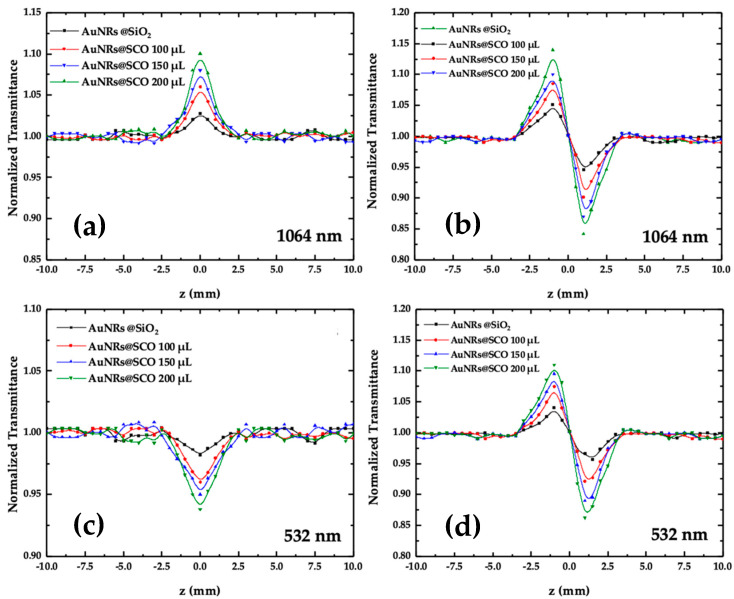
OA and CA Z−scans of AuNRs@SCO obtained under (**a**,**b**) 1064 and (**c**,**d**) 532 nm laser excitations.

**Figure 6 molecules-28-04200-f006:**
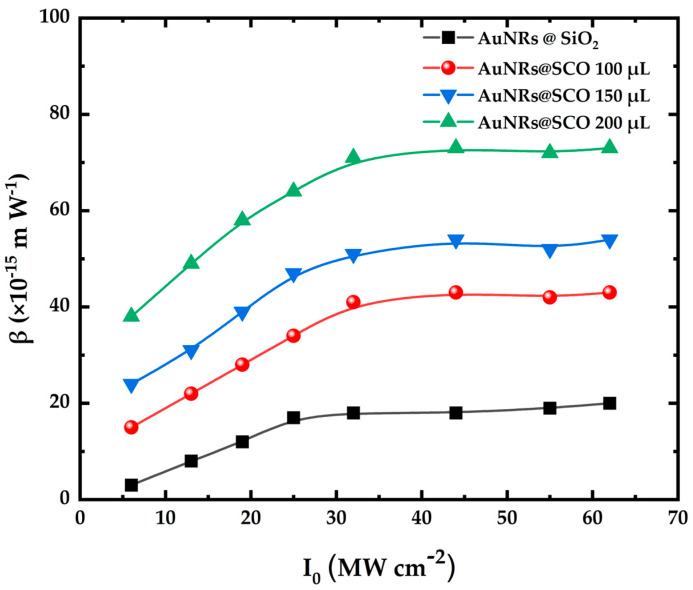
Dependence of nonlinear absorption coefficient (β) on incident laser intensity under 532 nm laser excitation for AuNRs@SCO.

**Figure 7 molecules-28-04200-f007:**
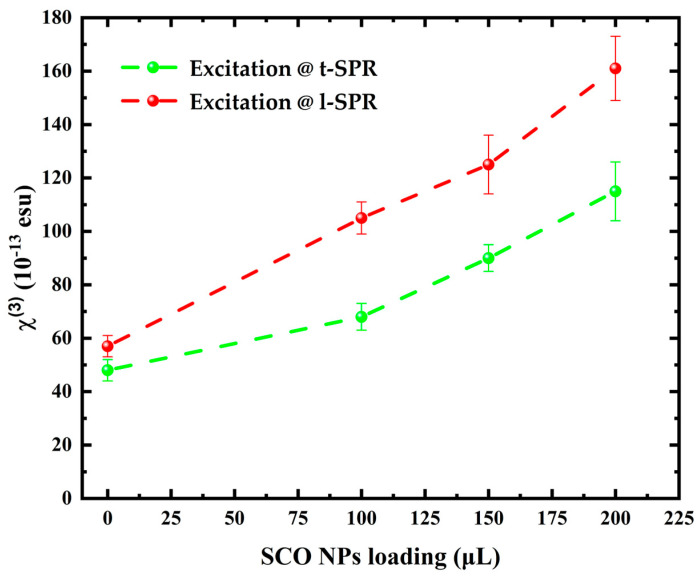
Dependence of the magnitude of the χ^(3)^ of AuNRs@SCOs, with the load of SCOs, under resonant excitation conditions at the t-SPR and l-SPR wavelengths (i.e., at 532 and 1064 nm, respectively).

**Figure 8 molecules-28-04200-f008:**
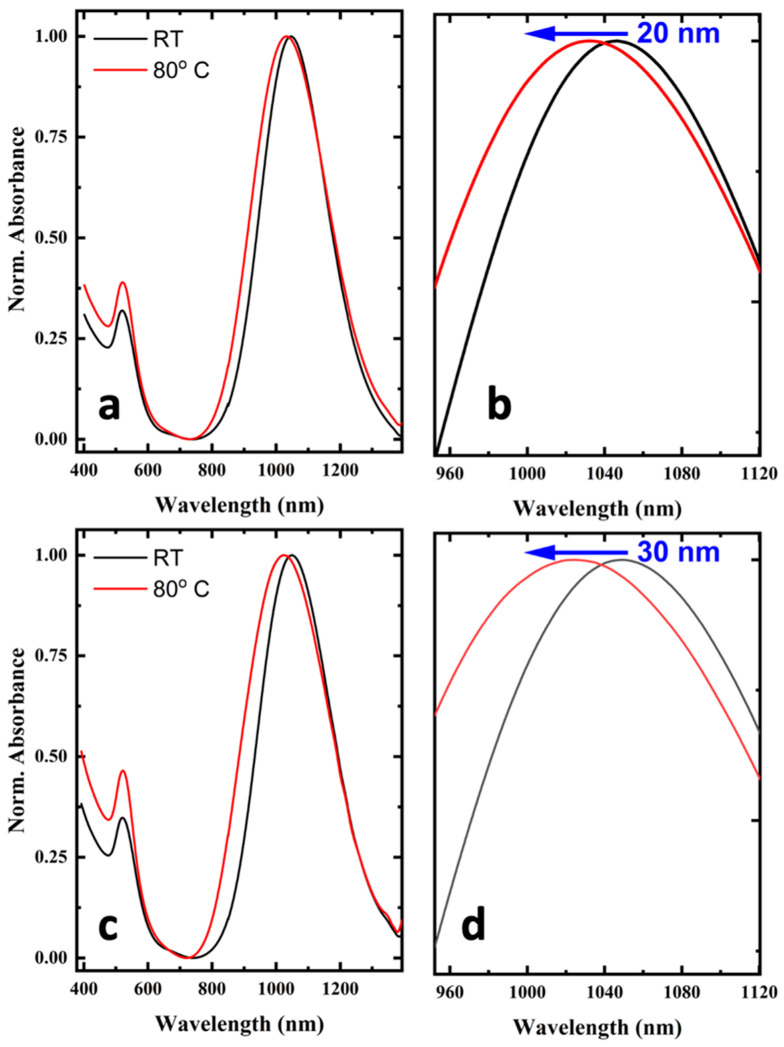
(**a**) Absorption spectra of AuNRs@SCO (loading 150 μL SCO) at RT (black line) and 80 °C (red line). (**b**) Magnification of the l-SPR area. See text for details. (**c**) Absorption spectra of AuNRs@SCO (loading 200 μL SCO) at RT (black line) and 80 °C (red line). (**d**) Magnification of the l-SPR area. See text for details.

**Table 1 molecules-28-04200-t001:** NLO parameters of AuNRs@SCOs under 532 and 1064 nm laser excitations.

	Excitation at 532 nm				Excitation at 1064 nm			
Sample	β (×10^−15^ m/W)	*γ*′ (×10^−18^ m^2^/W)	*n*_2_ (×10^−12^ esu)	χ^(3)^ (×10^−16^ esu)	β (×10^−11^ m/W)	*γ*′ (×10^−18^ m^2^/W)	*n*_2_ (×10^−12^ esu)	χ^(3)^ (×10^−13^ esu)
AuNRs	20 ± 4	−37 ± 3	−125 ± 10	48 ± 4	−68 ± 8	−80 ± 7	−272 ± 24	57 ± 4
AuNRs@SCO 100 μL	43 ± 4	−51 ± 6	−173 ± 14	68 ± 5	−86 ± 4	−125 ± 8	−425 ± 27	105 ± 6
AuNRs@SCO 150 μL	54 ± 5	−66 ± 4	−224 ± 20	90 ± 5	−118 ± 15	−152 ± 12	−516 ± 41	120 ± 11
AuNRs@SCO 200 μL	73 ± 8	−85 ± 4	−289 ± 20	115 ± 11	−130 ± 15	−190 ± 20	−646 ± 68	161 ± 12

## Data Availability

Not applicable.

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
