# Peer review of "Nanocomposites Based on Spin-Crossover Nanoparticles and Silica-Coated Gold Nanorods: A Nonlinear Optical Study"

_molecules, 2023, doi:10.3390/molecules28104200_

Round 1

Reviewer 1 Report

The manuscript " Nanocomposites Based on Spin Crossover Nanoparticles and Silica Coated Gold Nanorods: A Nonlinear Optical Study" submitted to Molecules devoted to the NLO studies of hybrid silica coated Au nanoparticles with the spin crossover nanoparticles of the 1-D iron(II) coordination polymer. The synthesis and possibility of applications of similar hybrids as thermally responsive MRI agents were recently published by this group in Dalt. Trans (https://doi.org/10.1039/D1DT02479E). This paper demonstrated how the 3rd order nonlinear optical properties of the hybrids AuNRs@SCO are dependent on the amount of SCO NPs grafted onto them. As a whole, the paper can be turned interesting for readers of inorganic chemistry division of Molecules. The manuscript is well written and well organized and can be recommended for publication in Molecules. As a comment, I recommend that the authors either change Figure 1 to be completely similar to the one in Dalt. Trans. or provide a reference to this article in the figure.

Author Response

We would like to thank the referee for the kind comments. Concerning Figure 1 we describe the experimental protocol for the SCO NP1 ( using the numbering of the paper in Dalton) while in the paper of Dalton we describe the experimental protocol of SCO NP2 ( using the numbering of Dalton’s paper). So, since there are differences between these two figures  ( and the relevant experimental protocols) we would like to keep the figure in its present form. In any case we added the reference in the Figure after the recommendation of the referee.

Reviewer 2 Report

This work offered by Tangoulis and co-workers is some interesting. In terms of the contents, this work could be published after fairly extensive modifications as followed:

(1). As for Fe(Htrz)2(trz)0.7(NH2trz)0.3](BF4)1.30.4SiO20.3H2O 0.1Acetone (SCO) and AuNRs@SiO2@SCO, gas adsorption studies of N2 and CO2 should be supplemented.

(2). The stability of Fe(Htrz)2(trz)0.7(NH2trz)0.3](BF4)1.30.4SiO20.3H2O 0.1Acetone (SCO) and AuNRs@SiO2@SCO in various organic solvents and water should be supplemented.

This work offered by Tangoulis and co-workers is some interesting. In terms of the contents, this work could be published after fairly extensive modifications as followed:

(1). As for Fe(Htrz)2(trz)0.7(NH2trz)0.3](BF4)1.30.4SiO20.3H2O 0.1Acetone (SCO) and AuNRs@SiO2@SCO, gas adsorption studies of N2 and CO2 should be supplemented.

(2). The stability of Fe(Htrz)2(trz)0.7(NH2trz)0.3](BF4)1.30.4SiO20.3H2O 0.1Acetone (SCO) and AuNRs@SiO2@SCO in various organic solvents and water should be supplemented.

Author Response

We would also like to thank Referee 2 for his/her kind comments. Concerning the solubility of the SCO NPs (water and EtOH only), we added a relevant sentence in the experimental section. For the AuNRs@SiO2@SCO it has been clearly stated in the experimental section that the dispersion is in DMF (only). The hybrids are stable only to this solvent.

Unfortunately, the N2, CO2 gas absorption studies are beyond the scope of this paper which is dedicated to the optical studies of these materials. Also, this technique is not available in the facilities of our university. Nevertheless, inspired by the comment of the referee we intend to investigate the possibility of performing these measurements in the near future for a paper related to more extensive physicochemical characterization of these systems.